# Autologous Blood Donation and Transfusion in Patients with Placental Malposition: A Single-Institution Pilot Study and Systematic Literature Review

**DOI:** 10.3390/healthcare12212138

**Published:** 2024-10-27

**Authors:** Iiji Koh, Kaoru Kawasaki, Kaori Moriuchi, Reona Shiro, Yoshie Yo, Noriomi Matsumura

**Affiliations:** Department of Obstetrics and Gynecology, Faculty of Medicine, Kindai University, 377-2 Ohnohigashi, Osakasayama 589-8511, Osaka, Japan

**Keywords:** autologous blood donation, autologous blood transfusion, low-lying placenta, placenta previa

## Abstract

Background: Autologous blood donation for placental malposition is common in Japan, but no studies have scientifically evaluated its usefulness. The purpose of this study was to evaluate the necessity for autologous blood donation for placental malposition. Methods: A retrospective study was conducted of patients who underwent autologous blood donation for placental malposition at Kindai University Hospital from 2012 to 2022. The primary outcome was the proportion of patients who were able to avoid allogeneic blood transfusion by autologous blood transfusion; secondary outcomes were autologous blood disposal rate, allogeneic blood transfusion rate, and complications of autologous blood donation and allogeneic blood transfusion. A systematic review of studies on autologous blood transfusion for placental malposition was conducted on PubMed. Results: Fifty-two patients (total placenta previa 16; marginal placenta previa 20; low-lying placenta 16) were included. Eight (15%) had complications at the time of autologous blood donation, including non-reassuring fetal heart rate, but no sequelae. Allogeneic blood transfusion was avoided by autologous blood transfusion in only five cases (9.6%). Autologous blood was discarded in nine cases (17%), seven of which had a low-lying placenta positioned normally at delivery. Allogeneic blood transfusion was performed in eight cases (15%) with no complications. In the systematic review, seven articles that met the inclusion criteria were selected for further evaluation. The results showed that there were no publications that scientifically demonstrated the benefit of autologous blood transfusion. Conclusions: The results of this study indicate that autologous blood donation for placental malposition has little benefit.

## 1. Introduction

Blood transfusion is essential to the safety of surgical procedures associated with heavy bleeding. Allogeneic blood transfusion carries the risk of complications, including post-transfusion graft-versus-host disease [1], which is particularly common and has a fatality rate in the Japanese population as high as 0.15%; transfusion-associated infections such as hepatitis and AIDS; and fever and urticaria due to antibodies to white blood cells, platelets, and plasma proteins produced by transfusion sensitization, the severity of which depends on the symptoms. Autologous blood donation is performed in obstetric and gynecological, cardiovascular, and orthopedic surgeries where massive blood loss is expected, to prevent complications associated with allogeneic blood transfusion [2].

Pregnant women with placental malposition have a higher risk of critical obstetric hemorrhage and are considered candidates for autologous blood donation. The Japanese Society of Autologous and Perioperative Blood Transfusion [3] and the Guidelines for the Practice of Obstetrics and Gynecology [4] recommend autologous blood donation for cases of placental malposition. However, predicting the amount of blood loss and the possibility of transfusion at delivery is difficult [5]. Autologous blood donation often leads to disposal and unnecessary blood transfusion [6]. Additionally, it must be performed under the supervision of many medical staff to prevent complications, including supine hypotensive syndrome, vagal reflexes, and fetal bradycardia, which raise Medicare costs [7]. The Practice Bulletin from the American College of Obstetricians and Gynecologists (ACOG) and the National Institute for Health and Care Excellence (NICE) do not mention autologous blood transfusion, and there is no international consensus. Some previous studies have concluded that autologous blood donation is beneficial based on the high autologous transfusion rate. However, a high autologous blood transfusion rate does not necessarily indicate that autologous blood donation is beneficial. In the present study, we aimed to evaluate whether autologous blood donation and transfusion is a truly useful treatment in modern obstetric care.

## 2. Materials and Methods

### 2.1. Patients

A single-center retrospective study was conducted from January 2012 to December 2022 at Kindai University Hospital. Singleton pregnant women who underwent autologous blood donation for placental malposition were included. Multiple pregnancies and cases with missing data were excluded. The Japanese guidelines for obstetrics and gynecology practice have no clear standards for the time and frequency of autologous blood donation for placental malposition [4]. In our institution, autologous blood donation is performed twice on average after approximately 34 weeks of gestation, similarly to previous reports [2,6]. In cases of total placenta previa, autologous blood is collected three times after 33 weeks of gestation to prepare for massive hemorrhage.

Using the ultrasound tomography method, the placental location was classified as total placenta previa (placenta completely covers the internal os), marginal placenta previa (the lower edge reaches the internal os but does not cover it), and low-lying placenta (the edge of the placenta is near to (<20 mm) but not overlying the os). The candidates for autologous blood donation were selected based on the standards of the Japanese Society of Autologous Blood Transfusion and Perioperative Blood Transfusion [3]. The present study was approved by the Ethical Review Board of Kindai University (approval number: R05-077).

### 2.2. Autologous Blood Donation

The patients’ data used for this study were collected from medical records. Three healthcare staff members (obstetrician, midwife, and nurse) observed the patients during autologous blood donation according to institutional policy. A non-stress test was performed 30 min before autologous blood donation and continued until 30 min after blood donation. Vessel puncture was performed using an 18-gauge needle, and extracellular fluid was administered before the blood donation to prevent complications, including supine hypotensive syndrome during blood donation. Blood pressure was measured three times during blood donation. Blood was collected and stored in a bag containing an adenine-added blood preservation solution (CPDA-1 solution) for 35 days. When the patients’ hemoglobin (Hb) levels were≥ 11 g/dL and 10–11 g/dL, 400 mL and 300 mL of blood, respectively, were collected. Autologous blood donation was canceled when the patient’s Hb level was <10 g/dL. The blood donation interval was at least 1 week, and no blood was collected within 3 days of the scheduled surgery.

### 2.3. Data Extraction

Data on subjects’ characteristics (age, parity, pregnancy outcomes); frequency, total volume, and complications of autologous blood donation; autologous blood discard rate; and allogeneic blood transfusion complications were extracted from their medical records.

The primary outcome was the proportion of patients who benefited from autologous blood transfusion, i.e., patients who were able to avoid allogeneic blood transfusion by autologous blood transfusion. The secondary outcomes were the autologous blood discard rate, the allogeneic blood transfusion rate, and complications of autologous blood transfusion and allogeneic blood transfusion. Cases of post-transfusion graft-versus-host disease, transfusion-related acute lung injury, and allergic reactions were evaluated as allogeneic blood transfusion complications. Hepatitis B, hepatitis C, and HIV infection tests were performed at 3 months post-operatively.

The criteria for identifying cases that truly required autologous blood transfusion were developed by the Japanese Clinical Guide for Critical Obstetrical Hemorrhage [8]. Patients were considered to be in true need of autologous blood transfusion if they had one or more of the following conditions: (1) persistent bleeding, (2) abnormal vital signs (oliguria, peripheral circulatory failure), (3) shock index of ≥1.5, (4) obstetric disseminated intravascular coagulation (DIC) score of ≥8, (5) fibrinogen level < 150 mg/dL, or (6) perioperative hemoglobin < 7 g/dL in the perioperative period.

### 2.4. Systematic Review of Autologous Blood Donation and Transfusion During Pregnancy

A systematic review was carried out, including article identification using the “PubMed” database, and hand searching, primary screening, and eligibility assessment according to the Preferred Reporting Items for Systematic Reviews and Meta-Analyses (PRISMA) guideline (Appendix A). The protocol of this review was not prospectively registered. A thorough search was conducted by two investigators on PubMed/MEDLINE to identify relevant articles on autologous blood donation and transfusion for patients with placental malposition. The strategy used combined Medical Subject Heading (MeSH) phrases based on the Boolean operators (“autologous blood collection” OR “autologous blood storage” OR “autologous blood transfusion” OR “autologous blood donation” OR “placenta previa”) AND (“placenta previa”). The initial search was performed on 1 September 2023 and updated on 16 February 2024. Observational studies and randomized control studies that investigated autologous blood donation and transfusion in patients with placental malposition were incorporated without imposing restrictions related to time period and geographic location. Articles that included only cases other than placental malposition, review articles, and articles published in languages other than English and Japanese were excluded. Those with incomplete or inaccessible information were also discarded. The Risk of Bias (ROB) of all included studies was examined using the Risk of Bias Assessment Tool for Nonrandomized Studies (RoBANS), which consists of six questions (Appendix A). The quality of the articles was evaluated using The Joanna Briggs Institute Meta-Analysis of Statistics Assessment and Review Instrument (The JBI-MAStARI). The evaluation considered several aspects: the study context, outcome and explanatory variables, specific inclusion criteria, measurement standards, topic description, and exact statistical analysis. The quality of the studies was classified as high (≥7 points), moderate (4 to 6 points), or low (<4 points) based on their score (Appendix A). The study outcomes were the autologous blood donation rate, the allogeneic blood transfusion rate, and the complication rate of the allogeneic blood transfusion rate.

### 2.5. Statistical Analysis

GraphPad Prism 6 was used for statistical analysis. Fisher’s exact test and the Mann–Whitney U-test were performed using GraphPad Prism 6 (Graph Pad Software, San Diego, CA, USA). *p* value < 0.05 was considered statistically significant.

The sample size for this study was determined using EZR (Saitama Medical Center, Jichi Medical University, Saitama, Japan), which is a graphical user interface for R (The R Foundation for Statistical Computing, Vienna, Austria) [9] as follows. The previous study showed that the autologous blood transfusion rate based on criteria including vital signs was approximately 40% [6,10]. By contrast, the autotransfusion rate in this study based on the criteria developed by the Japanese Clinical Guide for Critical Obstetrical Hemorrhage [8] was estimated to be approximately 20%. The necessary sample size was 53, assuming an α error of 0.05, a power of 80%.

## 3. Results

### 3.1. Subject Characteristics

Fifty-three women with a singleton pregnancy underwent autologous blood donation for placental malposition. Fifty-two cases were analyzed, excluding one case in which the blood-loss volume during delivery was unclear. The risk factors for placental malposition were prior cesarean section in eight cases (15%), uterine surgery in one (2%), and endometrial curettage in twelve (23%). The posterior wall was the most common placental site in forty-four (85%) (Table 1).

### 3.2. Timing and Amount of Autologous Blood Donation

Autologous blood was collected at a median gestational age of 34.3 (range, 30.3–37.1) weeks. Autologous blood donation tended to be performed earlier in patients with total placenta previa than in those with marginal placenta previa or a low-lying placenta. The median number of units and total blood volume were 2 (range, 1–4) and 750 (range, 200–1600) mL, respectively, with no significant differences according to the types of placental malposition (Table 2). The storage volume decreased due to anemia and complications during blood donation as noted in previous reports [11,12].

### 3.3. Complications of Autologous Blood Donation

Eight cases (15%) had complications during autologous blood donation, including hypotension in four cases (8%), non-reassuring fetal heart rate in one (2%), and both in three (6%) (Table 3). The non-reassuring fetal heart rate included only mild variables or late deceleration, and fetus status was subsequently confirmed as reassuring in all cases.

### 3.4. The Outcome of the 52 Cases That Underwent Autologous Blood Donation for Placental Malposition

The outcome of the 52 cases is shown in Figure 1A. Of the twenty patients who had marginal placenta previa at the start of autologous blood donation, two had a low-lying placenta and one had a normal-position placenta at delivery, and ten (63%) of the sixteen patients with a low-lying placenta had a normal-position placenta at delivery. Two patients with total placenta previa and two with marginal placenta previa required a cesarean hysterectomy, and nine of eleven patients with a normal-position placenta at delivery had a vaginal delivery. The need for blood transfusion was assessed as described in the Methods, and the 52 cases were classified as follows: (i) cases with allogeneic plus autologous transfusion (n = 8; total 6, marginal 2), (ii) cases requiring blood transfusion and undergoing autologous transfusion only (n = 5; total 1, marginal 2, low-lying; 2), (iii) cases without need for blood transfusion but with autologous blood transfusion (n = 30), and (iv) cases with discarded autologous blood (marginal 1, low-lying 8). In terms of avoiding allogeneic blood transfusions, group (ii) featured the cases that showed the benefit of autologous blood donation, but only 5 (9.6%) of the 52 cases were included in (ii). Blood loss in group (ii) was relatively homogeneous (1663–2648 g) and significantly higher than in group (iii) (386–1695 g) (*p* < 0.0001), suggesting the validity of the need assessment for blood transfusion in this study (Figure 1B).

### 3.5. A Systematic Review of Studies on Autologous Blood Transfusion for Placental Malposition

After applying the search method, 24 articles were identified, as described in the PRISMA flow diagram (Figure 2). A review of titles and abstracts was carried out, resulting in the selection of 15 articles for detailed full-text evaluation. Three articles that did not specify the patients’ obstetric condition and five articles with fewer than 10 participants were excluded. Seven articles that met the established inclusion criteria were selected for further evaluation in this systematic review. No randomized control study was found. A qualitative systematic review was conducted using observational studies.

Seven retrospective studies were included after primary and secondary screening. Three articles included only placental malposition [10,13,14] as in the present study, and four articles included topics other than placental malposition, such as any history of obstetric hemorrhage, multiple pregnancies, and pregnancies complicated by uterine fibroids [2,6,15,16]. The median rate of autologous blood transfusion was 72 (range, 32–92)% in previous reports, and it was also high in the present study at 83%. Two articles described the criteria for blood transfusion; the blood transfusion rate was low at 32% and 46%. The median rate of allogeneic blood transfusion was 9 (range, 4–18)%, and there were no cases of complications. All reports were descriptive, and no study scientifically demonstrated the benefit of autologous blood storage (Table 4).

## 4. Discussion

In our institution, an obstetrician, a midwife, and a nurse are present at the bedside, and a non-stress test is performed before, during, and after autologous blood donation, which allows for the early detection of an abnormal fetal heart rate. In previous studies [2,17], severe deceleration was included as a complication, whereas the present study included mild deceleration. In the present study, the complication rate (15%) of autologous blood donation (Table 3) was higher than that observed in previous studies (0–4%) [2,6,10,13,14,15,16]. Some studies did not monitor [18] or did not mention [19] the fetal heart rate during autologous blood donation, suggesting that the non-reassuring fetal heart rate may have been underestimated.

In the present study, the placenta in 63% of the cases of low-lying placenta migrated to a normal site, and the autologous blood discard rate was high in these cases. The transfusion rate was much higher in total placenta previa (7/16 cases, 44%). Both autologous and allogeneic transfusions were required in six of seven cases. Therefore, even in this high-risk population, autologous blood donation prevented only one case of allogeneic transfusion (Figure 1A). The placenta in 52% of low-lying placenta cases diagnosed at 30–33 weeks of gestation migrated to a normal site at delivery [20]. Therefore, the usefulness of autologous blood donation for low-lying placenta is low. However, the fact that autologous blood has been transfused without discard does not necessarily mean that autologous blood donation is useful. In this study, we developed criteria to assess the true need for blood transfusion according to the Japanese Clinical Guide for Critical Obstetric Hemorrhage. The actual number of subjects and the appropriate sample size described in the Methods section were comparable. The results showed that only 9.6% of cases were determined to have a true need for autologous blood transfusion while avoiding allogeneic blood transfusion. The rate of the true need for autologous blood transfusion did not differ among the types of placental malposition (Figure 1A). The higher complication rate with autologous blood donation (15%, Table 3) indicates that the benefits of autologous blood donation do not outweigh the risks to patients.

Six cases of placenta accreta were included in the present study. Three of six cases received both allogeneic and autologous blood transfusion. The remaining three cases received only autologous blood transfusion, which was considered unnecessary according to our criteria. None of the cases truly required autologous blood donation, although the number of the cases with placenta accreta was small. Large amounts of autologous blood donation are required to compensate for massive hemorrhage caused by placenta accreta, which can lead to preoperative anemia. Massive hemorrhage can also cause disseminated intravascular coagulation (DIC). Autologous blood is usually stored at room temperature, which inactivates coagulation factors. In cases with the possibility of placenta accreta, allogeneic blood transfusion, including sufficient fresh frozen plasma, should be prepared rather than large volumes of autologous blood at the time of caesarean section.

In the United States in the 1990s, autologous blood donation was reported to be less cost-effective because of the low allogeneic blood transfusion rate (0.6%, 13/2265 cases) and difficulty in predicting the possibility of transfusion at delivery [5,7]. It was also reported that although the blood transfusion rate for placental malposition is approximately 20% [12,21], autologous blood donation may be beneficial in only a few cases. Our systematic review of studies on autologous blood donation/transfusion for placental malposition showed no evidence to contradict the reports from the United States. There was heterogeneity in the autologous blood transfusion rates among the studies (Table 4). In the study where autologous blood was transfused when the vital signs were unstable or urinary output was decreased, the autologous blood transfusion rate was as low as 32% [6]. Few cases are likely to benefit from autologous blood transfusion, even in studies where the authors state that autologous blood donation is beneficial. The conclusions of the present study are similar to studies in the USA and European countries in the 1990s. However, the studies in the 1990s have been forgotten in Japan, and this has led to the current differences between Japanese and Western strategies for postpartum hemorrhage. With regard to Japanese healthcare, the fact that the studies cited are old does not mean that the theme of the present study is old. These old studies from the 1990s provide important evidence to inform change in the clinical strategy for placenta previa in the future.

The strength of the present study is that it evaluated the effectiveness of autologous blood transfusion in relation to the actual need for autologous blood transfusion. No other studies have evaluated the benefits of autologous blood transfusion in this way. Additionally, medical staff monitored maternal and fetal conditions closely, which allowed an accurate assessment of complications. However, limitations still exist. Firstly, we could not identify relevant indications to evaluate patient benefits. The potential benefits of autologous blood donation could be (i) to improve the rate of maternal mortality from massive hemorrhage and (ii) to avoid allogeneic transfusion-associated infections. In cases of maternal death, there occurs massive hemorrhaging that cannot be remediated by autotransfusion. And currently in Japan, HBV infections due to blood transfusion occur only once every few years in all diseases, and HIV and HCV infections have not been reported since 2014. Secondly, this is a retrospective study with a small number of cases and with no control group. At our institution, we have followed guideline recommendations and performed autologous blood donation in patients with placenta previa, except for those with anemia that prevents this procedure. Therefore, in our institution, placenta previa patients who do not undergo autologous blood donation cannot serve as a control group due to selection bias. In addition, there is no period when placenta previa was treated without autologous blood donation, and there is no historical control group. A multicenter study including institutions with a policy of not performing autologous blood donation for placenta previa was also not feasible. Under these circumstances, it is necessary to first examine in detail the cases in which autologous blood donation has been performed in order to investigate whether there is any potential benefit. Therefore, the present study does not have a control group. Thirdly, the obstetricians and anesthetists did not have uniform criteria for autologous blood transfusion. Prospective studies are needed to better evaluate the benefit of autologous blood transfusion in patients with placental malposition. In addition, autologous blood donation may remain an option in cases where allogeneic blood transfusion is difficult, such as in patients with rare blood types, to prepare for massive bleeding [11].

## 5. Conclusions

This study showed that autologous blood donation is not useful in cases of placental malposition. Although autologous blood donation is recommended in Japan, there is still no international consensus. Future studies are needed to scientifically evaluate the necessity of autologous blood donation.

## Figures and Tables

**Figure 1 healthcare-12-02138-f001:**
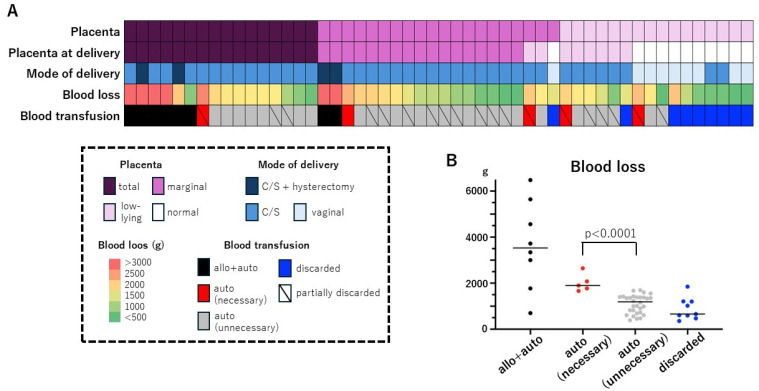
Outcome of 52 patients who underwent autologous blood donation for placental malposition. (**A**) Heat map presentation of individual cases. Placenta: position of placenta; total: total anterior placenta; marginal: marginal anterior placenta; low-lying: low-lying placenta. C/S: cesarean section; allo: allogeneic transfusion; auto: autologous transfusion; discarded: autologous blood was discarded. A shaded line in the box indicates that autologous blood was partially discarded. (**B**) Blood loss during delivery.

**Figure 2 healthcare-12-02138-f002:**
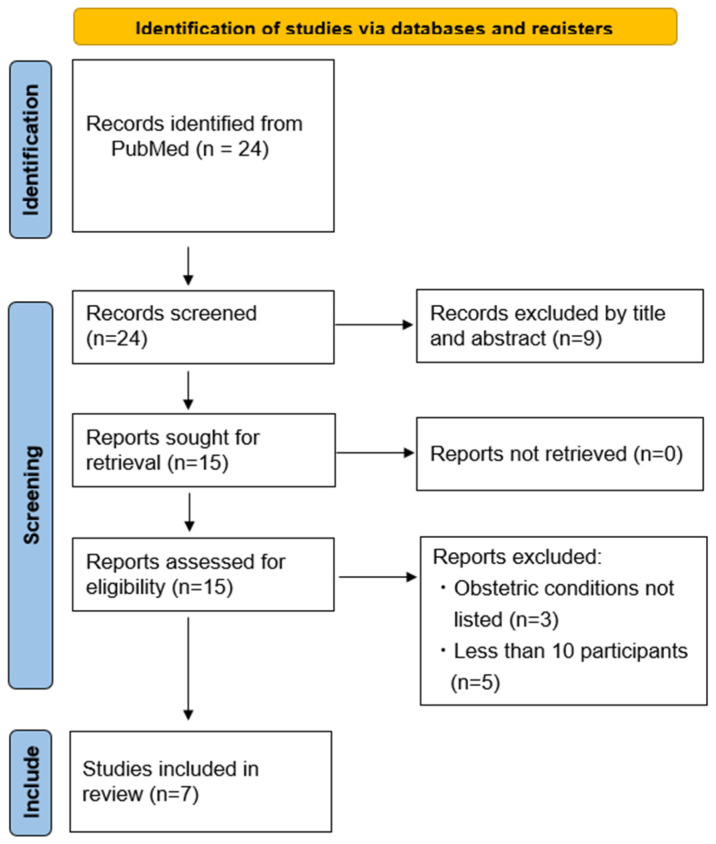
Selection process for studies according to the PRISMA flow diagram.

**Table 1 healthcare-12-02138-t001:** Subject characteristics.

	Median [Range], n (%)
maternal age	34 [25–43]
advanced maternal age	21 (40)
primipara	24 (46)
prior cesarean section	8 (15)
prior uterine surgery	1 (2)
prior endometrial curettage	12 (23)
placental site	
anterior wall	8 (15)
posterior wall	44 (85)

**Table 2 healthcare-12-02138-t002:** Timing and amount of autologous blood donation.

Types of Placental Malposition at Blood Donation (n)	First Blood Donation (Weeks)	Collected Blood	
		Units (U)	Total Volume (mL)
total cases (52)	34.3 [30.3–37.1]	2.0 [1–4]	750 [200–1600]
total placenta previa (16)	33.7 [30.3–36.4]	2.0 [1–4]	700 [200–1600]
marginal placenta previa (20)	34.1 [32.4–36.4]	2.2 [1–4]	860 [400–1600]
low-lying placenta (16)	35.3 [33.1–37.1]	1.7 [1–3]	650 [400–1200]

**Table 3 healthcare-12-02138-t003:** Complications of autologous blood donation.

	n (%)
complications	8 (15)
hypotension	4 (8)
non-reassuring fetal heart rate	1 (2)
both	3 (6)

**Table 4 healthcare-12-02138-t004:** A systematic review of studies on autologous blood transfusion for placental malposition. NA; not assessed.

Reference	Year	n	Autologous Blood Donation %	Allogeneic Blood Transfusion %	Complication of Allogeneic Blood Transfusion %	Criteria for Blood Transfusion
[15]	1988	11	82	18	NA	none
[16]	1994	10	90	NA	NA	none
[6]	2011	127	32	4	0	unstable vital signs or low urinary output
[2]	2014	56	91	7	0	none
[14]	2015	32	72	3	NA	none
[13]	2018	36	92	17	NA	none
[10]	2019	269	46	5	NA	intraoperative blood loss exceeded 1500 mL and further bleeding was expected or the hemoglobin level declined to <8 g/dL on post-operative days 0–1
Present study	2024	52	83	15	0	none

## Data Availability

The data that support the findings of this study are available from the corresponding author.

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
