# Peer review of "Autologous Blood Donation and Transfusion in Patients with Placental Malposition: A Single-Institution Pilot Study and Systematic Literature Review"

_healthcare, 2024, doi:10.3390/healthcare12212138_

Round 1
Reviewer 1 Report
Comments and Suggestions for Authors
The authors present their findings from a retrospective study on autologous blood donation and transfusion. The topic is of interest and relevant. The results are clearly presented. There is some inherent bias form the study design that is acceptable with retrospective trials. Some minor suggestions:
1. there is mention of a higher rate of complications from allogenic transfusion in the Japanese population - please provide clarifying data on what that rate is and how significant the reactions are
2. The study rakes place of a 10 year period 2012-2022. Any differences in overall transfusion rates, use of IV iron, Hgb values at time of delivery, and management of hemorrhage during that decade that may bias the results?
3. In the discussion I would highlight that in previa cases the rate of transfusion was much higher (44%) but that of the 7 patients requiring transfusion, 6/7 needed both autologous and allogenic transfusions. So even in this high-risk population, autologous blood donation only prevented 1 case of allogenic transfusion.
4. In the discussion, speculate about autologous blood donation in accreta cases. Given the close to 100% rate of transfusion in C-hys and accreta cases, one may make an argument for earlier and higher volume autologous donation for cases of high suspicion accreta diagnosed in the 2nd trimester. An argument against that would be the potential need for higher volume donation and complications that high volume donation could precipitate.
Reviewer 2 Report
Comments and Suggestions for Authors
The aim of the article presented is to evaluate the need for autologous blood donation for placental malposition and to assess whether autologous blood donation and transfusion is a really useful treatment in modern obstetric care (in this part it is necessary to define what is “a really useful treatment” in technical terms such as efficacy, efficiency, effectiveness, etc.), for which a retrospective study is performed to determine the frequency of complications such as hypotension and non-reassuring fetal heart rate after autologous blood transfusion.
There are two main points that concern me 1) the relevance and contribution of the study to scientific knowledge and 2) that the research design does not allow answering the research questions.
The chosen study design does not allow the study objectives to be answered, as it lacks a control group. In addition, the parameters measured do not allow us to conclude that the treatment is “useful”.
Other aspects to consider are:
The introduction contains little information on the relevance of the study.
The manuscript was presented as an article, however, its design contemplates a systematic review, given that systematic reviews have their own methodology, it is required to rely on some guidelines and to carry out the protocol of the review before carrying it out. If the authors did not contemplate these points, it is necessary to point out that a narrative review was performed and to remove that the review is systematic. However, this is the least of the study's drawbacks.
The results and experimental design do not support the conclusion of the study: “This study demonstrated that autologous blood donation is not useful in cases of placental malposition”, this due to the limitations of the design and that the patients who received autologous blood transfusion in the present study did not present any complications.
The references are very old, some dating back to 1990 and none in recent years, which shows that the object of the study is not a frontier subject.
Comments on the Quality of English LanguageMinor editing of English language required
Reviewer 3 Report
Comments and Suggestions for Authors
The manuscript describes the experience in a single institution in Japan about the use of autologous blood donation and transfusion in patients with placental malposition.
The study is well documented and all the necessary parameters have been studied and described.
The data presented in Table 4 evidence that, as in the present study, it’s quite difficult to assess the appropriatenes of the transfusion since the transfusion criteria are often not reported.
I would like to point out only that:
- in row 128 the bracket between “cases” and “total” must be removed;
- in rows 183 and 195 the word “transfusion” is used in place of “donation”
Reviewer 4 Report
Comments and Suggestions for Authors
As attached

Minor corrections needed
Round 2
Reviewer 2 Report
Comments and Suggestions for Authors
The authors have extensively justified the importance of the study, and the limitations of the study have been clarified in the manuscript.
Only one point remains to be completed. It is mentioned that the systematic review was based on PRISMA standards.
Therefore, it is necessary to fulfill the main requirements of a PRISMA-based systematic review, such as including the flow chart, including the checklist, describing the bias analysis, inclusion and exclusion criteria, protocol registration, etc.
Reviewer 4 Report
Comments and Suggestions for Authors
As attached

Round 3
Reviewer 2 Report
Comments and Suggestions for Authors
The authors have included the requirements of a systematic review based on PRISMA, so I have no other suggestions and consider that the manuscript can be published in its present form.